# Optimizing Niclosamide for Cancer Therapy: Improving Bioavailability via Structural Modification and Nanotechnology

**DOI:** 10.3390/cancers16203548

**Published:** 2024-10-21

**Authors:** Russell Wiggins, Jihoo Woo, Shizue Mito

**Affiliations:** Department of Medical Education, School of Medicine, University of Texas Rio Grande Valley, Edinburg, TX 78541, USA; russell.wiggins01@utrgv.edu (R.W.); jihoo.woo01@utrgv.edu (J.W.)

**Keywords:** niclosamide, anticancer drug, mechanism of action, bioavailability, drug delivery

## Abstract

Niclosamide, a drug originally used to treat parasitic infections, has shown potential as a possible cancer treatment, but poor absorptive and solubility properties are believed to limit its effectiveness. Researchers are exploring modified structures of the drug along with advanced delivery methods, particularly nanotechnology, to improve niclosamide’s absorption profile and improve its specificity in regard to targeting cancerous cells. These efforts intend to produce more effective tumor-targeting compounds, which can be employed as pharmacological cancer treatment options when appropriate. If successful, these efforts will provide new tools for scientists and doctors, offering a novel approach regarding the treatment of a variety of different cancers and contribute to the elucidation of oncobiology.

## 1. Introduction

Niclosamide (2′,5-dichloro-4′-nitrosalicylanilide) was initially proposed as an anthelmintic agent. In recent years, the anti-neoplastic properties of the drug have drawn the attention of cancer researchers for potential use as a repurposed pharmacologic agent to be employed as a mono or adjunct therapy in the setting of malignancy [1]. Recent studies showed evidence that niclosamide indeed has multiple anti-neoplastic properties, eliciting effects in many major proto-oncogenic and tumor-suppressing cellular pathways, affecting processes such as the signal transducer and activator of transcription 3 (STAT3), nuclear factor-kappa B (NF-κB), wingless-related integration site (Wnt)/β-catenin, mammalian target of rapamycin (mTOR), and others [2,3,4]. Niclosamide was also shown to disrupt metabolic processes through its action as a mitochondrial uncoupling agent, and through inhibitory effects on aerobic glycolysis (the Warburg effect) in certain cancer lineages in vitro and in vivo [5]. These pathways and processes are often involved in initiating and regulating cellular growth, maintaining homeostasis, modulating metabolism, contributing to mitotic division, promoting angiogenesis, and facilitating cancer metastasis. Niclosamide’s ability to interfere in several key cellular pathways in such a manner as to antagonize the unmitigated growth and spread exhibited by cancerous cells prospectively makes it a versatile compound to be utilized in pharmacological or combined cancer treatment protocols [6]. Despite the elucidation of niclosamide’s promising mechanisms of action across several different cancer lineages, challenges related to its bioavailability (*F*) persist. These challenges arise in part due to (1) poor solubility resulting from the hydrophobic nature of its chemical structure, which consists of aromatic rings and balanced polar groups, and (2) its rapid degradation by first-pass metabolism in the liver and intestine via cytochrome P450 enzyme-mediated hydroxylation and UDP-glucuronosyltransferase (UGT) enzyme-mediated glucuronidation [7]. These issues may be addressed through the discovery of niclosamide derivatives that enhance solubility and inhibit first-pass metabolism by hindering enzymatic binding and the formation of hydroxylated niclosamide, niclosamide glucuronides, and niclosamide sulfate, all while maintaining the key chemical properties necessary for the desired antitumor activity (Table 1) [7,8].

Alternatively, novel methods for drug delivery may allow for the conservation of niclosamide’s structure and circumvent the hurdle associated with *F* and solubility. This systematic literature review seeks to compile and analyze the contemporary research on niclosamide’s structural characteristics that contribute to its action as an uncoupler, its relation to other experimental compounds, malignancy, key cancer-related pathways, potential and current oncologic applications, the development of derivatives, preclinical effectiveness, and possible future clinical applications.

## 2. Niclosamide’s Mechanisms of Actions

### 2.1. Mitochondrial Dysfunction

Mitochondria harness potential energy created via an electrochemical gradient between the mitochondrial matrix and the intermembrane space called the proton motive force (Δ*p*) to produce ATP [13]. Hydrogen atoms/protons are transferred across the inner membrane from the mitochondrial matrix during oxidative phosphorylation (OXPHOS) [13,14]. ATP production depends on maintaining Δ*p* and channeling this electrochemical gradient through choke points containing ATP synthase [13,15]. Δ*p* can be dissipated by numerous factors including electrolyte imbalance, physiologic response, and chemical insult from “uncoupling agents” [16,17]. Mitochondrial uncoupling agents dissipate the proton gradient by allowing protons to flow back into the mitochondrial matrix without passing through ATP synthase, the gradient energy is instead dissipated as a form of heat, invoking metabolic stress that often leads to cell death [18]. Although the anticancer activity of niclosamide involves multiple signaling pathways, mitochondrial membrane depolarization by niclosamide is believed to lead to mitochondrial matrix condensation and subsequent cytochrome *c* (Cyt*c*) release, which initiates apoptosis (Figure 1).

Niclosamide has been shown to act as an uncoupling agent. Specifically, niclosamide acts as a protonophore, an electrogenic transporter of protons across lipid membranes [22]. This disruption of OXPHOS causes increased activity of the electron transport chain (ETC), wherein incomplete reduction in electrons transpires at cytochrome complexes I and III, leading to the generation of reactive oxygen species (ROS) [6,23]. These potent oxidants reach intracellular structures, causing damage to proteins, lipids, DNA, and eventually, cell death [2]. This uncoupling effect leads to mitochondrial dysfunction, resulting in energy depletion, ROS-induced stress responses, and activation of AMP-activated protein kinase (AMPK) [5,13]. AMPK is activated in response to ATP depletion and inhibits anabolic activity, promotes autophagy, cell cycle arrest, and apoptosis [20,24]

The pro-apoptotic effect of niclosamide extends beyond its interference with mitochondrial ATP generation. Research indicates that niclosamide’s mitochondrial membrane potential disruption also results in mitochondrial depolarization, mitochondrial matrix condensation, and the release of Cyt*c* into the cytosol [25,26]. The release of Cyt*c* is a key initial step in the intrinsic apoptotic pathway, activating caspases to execute cell demolition through the activation of various degradation enzymes [27]. By inducing Cyt*c* release, niclosamide activates the apoptotic cascade, driving susceptible cancer cells toward programmed cell death [28].

These niclosamide–mitochondria interactions offer potential avenues for targeting susceptible cancer lineages. Niclosamide, with its combined effects of energy depletion, induction of oxidative stress, and initiation of the intrinsic apoptotic pathway, makes it a powerful, mechanistically diverse antitumor agent in susceptible cancers [26].

### 2.2. STAT3 Signaling Pathway

The Signal Transducer and Activator of Transcription 3 (STAT3) signaling pathway plays an important role in tumorigenesis, immune evasion, and apoptotic resistance [29]. STAT3 is a member of the STAT family of transcription factors that participate in cell growth, differentiation, and survival after cytokine and growth factor-mediated signal transduction [29]. Once STAT3 is activated, it forms a dimer complex and translocates to the nucleus, promoting the expression of genes that contribute to cellular proliferation and angiogenesis (Figure 2) [30]. Inhibition of the STAT3 signaling cascade has been shown to impede the proliferation and metastasis of certain cancers [31].

### 2.3. mTOR Pathway

mTOR (mechanistic target of rapamycin) is a key pathway responsible for promoting cellular growth and survival while inhibiting cellular autophagy when activated. It exists as two distinct complexes, mTORC1 and mTORC2 [33]. Under normal physiologic circumstances, it reacts to growth factors, energy status, and cytokines and is downstream from regulators like phosphoinositide 3-kinases (PI3K), protein kinase B (Akt), and phosphatase and tensin homolog (PTEN) [33]. Mutations in these regulators lead to constitutive activation of the mTOR pathway, causing a variety of cancers. Therefore, mTORC1 is the primary target in pharmacologic-based interventions for cancer therapy, but resistance has been known to develop.

Compensation is one such suggested mechanism of tumor resistance specific to mTORC1 targeting, wherein the inhibition mTORC1 normally exerts on the PI3K/Akt pathway is countered by pharmacological intervention with mTORC1 inhibitors, paradoxically leading to overactivation within the pathway [34]. Such a mechanism suggests that dual targeting of mTORC1/mTORC2 kinases (which activate the PI3K/Akt kinases) may prove to be a more effective and efficient strategy to combat tumor proliferation and metastasis [35]. Dual mTORC complex targeting may be a strategy of necessity in cases of gained compensatory resistance, as seen with traditional mTOR inhibitors.

However, niclosamide has been known to only disrupt mTORC1 [36]. It is believed to disrupt mTOR signaling through activation of TSC which inhibits mTOR1 activity (Figure 2) [37]. Therefore, current research suggests using niclosamide as a supplement to Paclitaxel due to limited mTORC1 targeting [38].

### 2.4. Wnt/β-Catenin Pathway Inhibition

The Wnt/β-catenin pathway is an essential regulator of cellular proliferation, differentiation, and homeostasis [39]. Signal transduction is initiated by the interaction of Wnt ligands with Frizzled serpentine receptors and their low-density lipoprotein co-receptors (LRP5/6), which act on the downstream effector protein β-catenin [40]. When activated by Wnt signaling, β-catenin acts as a signal transducer, associating with T-cell factor (TCF) and lymphoid enhancer factor (LEF), to travel to the nucleus and upregulate transcription of an array of genes, including several proto-oncogenic (cyclin D1, cellular myelocytomatosis (c-Myc), vascular endothelial growth factor (VEGF), and matrix metalloproteinase 7 (MMP7), Axin2, etc.) [40]. In the absence of Wnt signaling, β-catenin is continuously degraded by a multi-protein β-catenin destruction complex [40]. This destruction complex facilitates a series of phosphorylation events along serine–threonine residues; in particular, motifs are found on the β-catenin protein so that it may be tagged for ubiquitination and subsequent proteasomal degradation, effectively downregulating β-catenin target gene transcription [41].

Niclosamide has been shown to disrupt Wnt/β-catenin signaling by several different mechanisms [42]. For instance, niclosamide led to the degradation of LRP6, one of the essential co-receptors of the Frizzled serpentine receptor, inhibiting the initial signal transduction at the level of ligand/receptor interaction in ovarian cancer (Figure 2) [42]. It was also observed that the IR-induced Wnt/β-catenin apoptosis of tumor cells was increased in niclosamide-treated breast cancer MDA-MB-231 samples in vitro [43].

### 2.5. NF-κB Pathway

The nuclear factor kappa B (NF-κB) signaling pathway is an important regulator of many cellular processes, including inflammation and the survival of both cancer and immune cells [44]. NF-κB is normally present as an inactive, cytoplasmic form in complex with an inhibitor protein (IκB) which blocks nuclear translocation [44]. However, upon exposure to pro-inflammatory stimuli or other activating signals, IκB kinase (IKK) is activated, and phosphorylates IκB, promoting its degradation [45]. The degradation event releases NF-κB dimers p65 (RelA) and p50, which translocate to the nucleus and induce transcription of genes in support of inflammation, cell proliferation, and survival [45]. These processes can be often deregulated in cancer cells [45].

Niclosamide recently emerged as a potent inhibitor of the NF-κB pathway [46]. Niclosamide uniquely inhibits IKK, one of the key kinases involved in the activation of the NF-κB pathway, by repressing IκB degradation (Figure 2) [46,47]. Through this action, IκB remains intact within the cytoplasm, blocking NF-κB from translocating to the nucleus [46]. As a result of IκB retention, expression of NF-κB-dependent, anti-apoptotic genes decreases, thus sensitizing cancer cells towards features of apoptosis, potentially improving therapeutic outcomes [46].

## 3. Brief Comparison of Niclosamide with Other Cancer Therapeutics Targeting Similar Pathways

### 3.1. Niclosamide v. CCCP and FCCP (Mitochondrial Uncoupling)

Carbonyl cyanide *m*-chlorophenylhydrazone (CCCP) and carbonyl cyanide-*p*-trifluoromethoxyphenylhydrazone (FCCP) are protonophores utilized in the laboratory setting to inhibit oxidative phosphorylation [48]. They share the same mechanism in which they affect the proton motive force (Δ*p)* and the subsequent consequences of Δ*p* disruption with niclosamide, namely a reduction in ATP synthesis, generation of ROS, initiation of the metabolic stress response, and eventual apoptosis [17,49]. Although effective uncouplers, CCCP and FCCP have shown to be either ineffective or require higher concentration compared to niclosamide in certain cancer cell lineages (FCCP: IC_50_ (μM) = 6.6 in Calu-6 cells after 72 h, CCCP alone did not cause apoptosis in 143B TK^−^, SKW6, MCF-7, Jurkat, and CEM cells but after extended exposure, caused apoptosis in FL5.12 and Jurkat cells) [50,51,52,53].

Niclosamide also has unique characteristics compared to the other protonophores, which favor its utilization as a neoplastic agent. Niclosamide demonstrates selectivity, wherein uncoupling action and ROS production preferentially favor cancer cells over normal cells [54]. For instance, niclosamide displayed tumor selectivity and spared normal human bone marrow and fibroblast cells while targeting AML cells in vitro, possibly due to unique tumor membrane properties, metabolic characteristics, or higher rates of drug uptake [46].

In addition to its role as an uncoupler, niclosamide’s ability to target multiple signaling pathways pronounces its anticancer activity over other mitochondrial uncouplers (Figure 2) [11,32]. Niclosamide can induce mitochondrial uncoupling together with the inhibition of cancer-specific pathways, allowing for broader coverage and more targets of opportunity within cancerous cells [3,32].

### 3.2. Niclosamide v. WP1006 (STAT3 Inhibitor)

The experimental STAT3 inhibitor WP1066 has demonstrated preclinical and clinical efficacy through inhibition of phosphorylation of STAT3, reducing target gene transcription and inducing apoptosis in cancer cells [55,56,57]. Despite WP1066’s efficacy in disrupting the STAT3 pathway, it showed limitation as a standalone treatment as immune-effector responses, TNF-α, IFN-γ and IL-2 were not improved in a first-in-human Phase I trial, and a relatively high IC_50_ compared to niclosamide in STAT3 HeLa cells (IC_50_: WP1066 = 2.43; niclosamide = 0.25 μM), revealing a need for other efficacious agents with higher degrees of specificity [34,35,58].

There is evidence that niclosamide has inhibitory effects on the STAT3 signaling pathway [58]. Niclosamide blocks the phosphorylation of STAT3 at tyrosine 705, which is critical for STAT3 activation and nuclear localization [58], thereby blocking STAT3 from binding to DNA and activating transcription of genes involved in survival and proliferation (Figure 2) [59]. There is also evidence of modulation of the IL-6/STAT3 axis, wherein niclosamide mitigates signal transduction at the level of the JAK kinase [59,60].

### 3.3. Niclosamide v. IKK Inhibitors (NF-kB Inhibition)

Bortezomib is an FDA-approved proteasome inhibitor that inhibits the degradation of IκB, thus keeping NF-κB in the cytoplasm [61]. Although bortezomib is effective against many malignancies, its clinical use is impeded by side effects such as peripheral neuropathy and the development of drug resistance [62,63]. IKK inhibitors such as ML120B target IKK specifically but often exhibit off-target effects, as other cytoplasmic and nuclear proteins that are not associated with NF-kB are substrates of IKK [64]. Conversely, niclosamide’s targeting of IKK may produce these off-target effects as well [61,64].

## 4. Optimizing Niclosamide’s Bioavailability

### 4.1. Structural Optimization of Niclosamide for Enhanced Uncoupling Activity

To address niclosamide’s unfavorable *F* profile, researchers explored the feasibility of synthesizing derivatives with improved pharmacokinetics without compromising its efficacy as an anti-neoplastic agent. Recent studies indicated that derivatives, such as niclosamide ethanolamine (NEN) and niclosamide piperazine (NPP), among others, may offer enhanced solubility [65,66]. Understanding the structure–function relationship of these functional groups is essential for designing novel derivatives with improved efficacy and safety profiles.

An important rule of thumb when developing niclosamide derivatives is to maintain the chemical properties of functional groups and moieties responsible for inducing mitochondrial uncoupling activity. A study conducted by Terada in 1990 showed the following characteristics are required to act as an uncoupler (Figure 3) [67]:**An acid-dissociable group:** An acid-dissociable group on an uncoupler allows it to alternate between a protonated (neutral) form and a deprotonated (negatively charged) form. Once inside the matrix, where the pH is relatively higher (less acidic) than in the intermembrane space, the acid-dissociable group releases the proton (H^+^). This action converts the uncoupler into its deprotonated, negatively charged form. The study notes a weak acidic group, such as a phenolic hydroxyl (OH), or a carboxylic acid (COOH), may serve as one.**A large hydrophobic moiety:** A large hydrophobic group is necessary for integrating the uncoupler into the lipid bilayer of the mitochondrial membrane. This moiety enhances the lipophilicity of the uncoupler, allowing it to reside within the hydrophobic environment of the membrane. It helps stabilize the uncoupler within the membrane, ensuring effective interaction with the lipid bilayer and facilitating its function in transporting protons across the membrane. Examples are tert-butyl groups or long alkyl chains.**A strong electron-withdrawing group:** The presence of a strong electron-withdrawing group is essential to stabilize the negative charge that forms on the uncoupler when it dissociates to release a proton. This stabilization is important for maintaining the uncoupler in a form that can continue to shuttle protons across the membrane. Electron-withdrawing groups increase the acidity of the dissociable group, making it easier for the uncoupler to release a proton and participate in the proton transport process. Functional groups like nitro (NO_2_), trifluoromethyl (CF_3_), or cyano (CN) may fulfill this role.

Additionally, for weakly acidic uncouplers, the rule of thumb according to Terada is as follows [67]:Uncoupling is due to its action as protonophores (allowing protons to flow freely across the membrane without being coupled to ATP synthesis).The stability of uncoupler anions in the hydrophobic membrane is crucial.High stability is achieved through mechanisms specific to each uncoupler, involving delocalization of the polar ionic charge.

Terada proposes that for optimal action as an uncoupling agent, the stoichiometry should be smaller, and the number of “cycles” (a “cycle” refers to the process in which one molecule of the uncoupler carries one proton from the cytosolic phase to the matrix space [67]) should be greater to compensate for the loss of uncouplers, as some uncoupler molecules could be bound by membrane components instead of target phospholipids.

Another challenge is that an uncoupler must have a high turnover rate because the total number of protons pumped by the ETC is higher than the concentration of an uncoupler [67]. This means an uncoupler must continuously and quickly pick up protons across the membrane [67]. For instance, with succinate as the respiratory substrate (which donates electrons to the ETC), an uncoupler must pump back approximately 400 cycles per second to induce full uncoupling [67]. This indicates a substoichiometric relationship between (1) the uncoupler molecule and (2) the number of protons pumped. This relationship determines what acts as an effective uncoupler, as it should be involved in multiple cycles per oxygen atom consumed during respiration [67].

### 4.2. Overcoming Niclosamide’s First-Pass Metabolism Challenges

Another factor that can contribute to the low bioavailability of a drug, other than low water solubility, is its metabolism. When orally ingested, drugs enter the gastrointestinal tract, where they are absorbed into the portal circulation and metabolized by the liver before reaching the systemic circulation. Controlling this factor could improve oral niclosamide bioavailability; hence, few researchers have delved into unraveling the mechanism of niclosamide’s first-pass metabolism. A study identified key metabolic enzymes as the cytochrome P450 enzyme, specifically CYP1A2, and UDP-glucuronosyltransferase (UGT1A1) located in the liver and intestine [68].

Expanding on this discovery, an in vivo study examined the relative contributions of P450 and UGT enzymes to the degradation of niclosamide. Notably, UGT enzymes, particularly those located in the intestine, played a more significant role than those in the liver [7]. Although intestinal UGT enzymes typically have lower Km values compared to hepatic UGT, they exhibit a higher Vmax (metabolic rate) at elevated concentrations of niclosamide. This suggests that under conditions of high substrate availability, intestinal UGT enzymes are more active than their hepatic counterparts [7].

Another important observation from this study is that UGT enzymes have a compensatory upregulation system. In mice without the hepatic cytochrome P450 enzyme, particularly CYP1A2, there was a significant increase in hepatic UGT1A1 mRNA and protein expression, showing that UGT has a compensatory mechanism for CYP metabolism [7].

The key takeaway from this study is that to increase the bioavailability of niclosamide by inhibiting its degradation, both UGT and CYP450 enzymes should be inhibited, rather than just inhibiting CYP450 alone, due to its compensatory mechanism [7]. However, there are no additional in vivo studies on niclosamide with inhibition of these enzymes, leaving room for more research.

## 5. Niclosamide with Modified Structures

### 5.1. Niclosamide Ethanolamine Salt

A recent study showed that niclosamide ethanolamine (NEN) demonstrated better efficacy than niclosamide in both in vitro and in vivo metastatic colon cancer models with HCC cells; NEN and niclosamide both demonstrated specificity in targeting HCC cells over normal cells at least seven-fold more, and oral administration of NEN showed a greater reduction in growth of both genetically induced liver tumors and patient-derived xenografts than niclosamide, coinciding with its better solubility and bioavailability due to its improved pharmacokinetic profile [65]. In another recent study, NEN was shown to significantly diminish hepatic metastasis in colon cancer models and intestinal polyp in APC min/+ mice and that oral NEN in mice induced targeted mitochondrial uncoupling in the liver [5]. The data may indicate that the increased potency of NEN is closely attributed to pharmacokinetic enhancements that improved solubility in aqueous environments and enhanced cellular uptake, thus improving availability at the target site [5,65,69].

The mechanism behind the improved *F* of NEN lies in its ethanolamine functional group, making this prodrug more hydrophilic compared to the parent compound. The ethanolamine group contains both hydroxyl (OH) and amine (NH_2_) moiety groups [65]. The presence of strongly electronegative atoms within the functional groups allows for hydrogen bonding with water molecules when suspended in an aqueous solution. Therefore, the incorporation of these moieties makes NEN more hydrophilic and soluble in water-based fluids than niclosamide. The increase in aqueous solubility is important, as it allows for aqueous diffusion to transpire, meaning the drug may be delivered via circulation and deposited into target tissues in aqueous compartments [70].

NEN also maintains substantial hydrophobicity through the preservation of aromatic rings and other hydrophobic moieties homologous to niclosamide. These hydrophobic segments allow for compound permeability in lipid-rich environments (including cellular membranes), a characteristic perhaps more important, as the body contains numerous lipid barriers [70]. The amphipathic characteristics of NEN are pivotal in contributing to drug solubility, deposition in target tissues, and permeability.

### 5.2. Niclosamide Piperazine Salt

NPP is a piperazine salt form of niclosamide, also known as Taenifungin, an anthelmintic drug that demonstrated efficacy and safety in vivo in sheep and cattle for the treatment of Hymenolepis Fraterna [71]. Notably, NPP has better water solubility (0.023 mg/mL at 37 °C) compared to niclosamide (0.003 mg/mL at 37 °C) [10]. The enhanced pharmacokinetic profile of niclosamide piperazine (NPP) may be attributed to its piperazine ring at the aniline nitrogen site of niclosamide, which replaces the hydrogen at the amide moiety. This introduction of the piperazine ring may improve hydrophilicity through its additional nitrogen atoms, facilitating hydrogen bond formation and enhancing solubility in aqueous environments and absorption.

However, NPP’s resistance to metabolism by liver enzymes in vivo and in vitro has yet to be fully explored. Niclosamide is primarily metabolized via CYP450 and glucuronidation, involving the binding of its hydroxyl, amine, and carboxyl groups [7]. The nitrogen atoms in the piperazine ring may create steric hindrance at the CYP450 and glucuronidation binding sites, potentially increasing resistance to hydrolysis and oxidative degradation, thus improving bioavailability.

Moreover, NPP’s anti-cancer efficacy has not been extensively studied. One investigation showed NPP’s efficacy as an uncoupler is comparable to that of NEN [64]. In this study, administration of NPP resulted in reduced high-fat diet-induced obesity, hepatic steatosis, and hyperglycemia in mice through its hepatic mitochondrial uncoupling effect similar to NEN [64]. Another study reported that NPP exhibited anti-cancer activity against glioblastoma cells in vitro, with an IC_50_ value of 1.50 µM, while NEN had an IC_50_ value of 1.834 µM, which is relatively similar to the IC_50_ value of niclosamide [72]. With its promising potential still under-researched, NPP shows promise as a better candidate compared to niclosamide.

### 5.3. Niclosamide Derivatives and Their Structure-Activity Relationship (SAR) Studies

Recently, researchers focused on increasing mitochondrial uncoupling activity using niclosamide derivatives, adhering to the structural motifs and outline proposed by Terada [67]. For instance, a structure–activity relationship (SAR) study regarding niclosamide derivatives in human glioblastoma U-87 MG cells was conducted (Table 2) [73]. Among them, Table 2 compounds **6** (with R^4^-N_3_) and 7 (with R^4^-CF_3_) followed the proposed motifs of Terada, as they were able to replace R^4^ functional group from NO_2_ to N_3_ or CF_3_, and in turn showed enhanced integration into mitochondrial membranes and conferred stability of uncoupler anions. The study claims that the phenol OH group at R^1^ has increased antineoplastic activity, while the chloride group Cl at R^2^ showed no significant effect (compounds **2** and **13**). In addition, Table 2, compound 5 (R^3^–H, R^4^-NH_2_) reduced the efficacy of anti-cancer action, which is consistent with the above-mentioned property of an uncoupler.

However, the degree to which prospective niclosamide derivatives meet Terada’s criteria is not clear in other studies. Despite Terada’s requirement of having acid dissociable groups, other derivatives from a separate study, (Table 3, compounds **10** (R^1^-(CH_2_)_5_NH) and compound **11**, (R^1^-CH_3_CH_2_NH_2_)) showed substantial uncoupling activity [74]. The alkyl-O-amino functional groups that replaced the acid dissociable group (OH) of niclosamide are base or neutral, which is contradictory to Tarada’s claim. Yet, it should be noted that both Table 3, compounds **10** and **11** have excellent aqueous solubility. Particularly, compound **11** is about 3300-fold more water-soluble compared to niclosamide, and it has great potential for cancer therapy.

An additional notable SAR study by He et al. reported the efficacy of a niclosamide derivative in the in vitro inhibition of a small cell lung cancer (SCLC) lineage (Table 4) [75]. They modified the 2-free hydroxyl functional group, substituting degradable and non-degradable esters (Table 4, compound **2**, R^1^-COCH_3_), which resulted in a mildly reduced IC_50_ in comparison to niclosamide, and total attenuation of its anti-tumor action in the non-degradable ester derivatives. They believed this demonstrated the importance of the 2-free hydroxyl functional group in the targeting of SCLC.

Another structure–activity relationship (SAR) study utilized the TOPflash assay to evaluate the inhibitory characteristic of niclosamide on the Wnt/β-catenin pathway (Table 5) [76]. The TOPflash assay is a reporter gene assay that measures the transcriptional activity of β-catenin, providing insights into the activation of the Wnt signaling pathway. The study indicated that methylation of the hydroxyl group at R^1^ or substituting R^1^ with hydrogen led to decreased activity in Wnt/β-catenin signaling. In contrast, replacing the chloro group at R^3^ with a methoxy group (see Table 5, compound **7**) demonstrated similar potency to niclosamide, while substituting R^3^ with hydrogen resulted in only a slight reduction in potency. These findings suggest that modification of R^3^ is feasible without compromising drug efficacy, potentially allowing for R^3^-PEG substitutions to attach reagents for cancer targeting.

These studies suggest derivatives may be effective in some pathways, while ineffective in others, as certain functional groups seem to play key roles within certain pathways. One may also infer from these findings that a derivative’s anti-neoplastic effectiveness may be entirely dependent on which pathways are active/inactive within cancerous cells.

Finally, another novel strategy that can be employed to gain improvement in the solubilization, stability, and delivery of niclosamide is nanoparticle-based delivery systems.

## 6. Nanoparticle-Based Delivery Systems for Niclosamide

### 6.1. Targeting

Nanoparticle-based delivery systems are a collection of engineered vehicles comprised of various < 1000 nm substances that aim to improve solubility, *F*, confer stability, reduce toxicity, and facilitate targeted delivery [77]. These nanotechnologies allow for a staggering level of creativity and modification. A variety of proteins, functional groups, and organic molecules may be incorporated into the structure of the vehicles to serve as immunomodulators, act as ligands, and alter pharmacokinetics [78].

Utilizing these innovative methods, niclosamide was delivered both in vitro and in vivo through a range of nanotechnology platforms, including nanocrystals, polymeric nanoparticles, lipid nanoparticles, nanofibers, carbon nanoparticles, and micelles [79].

In an effort to enhance the efficacy and targeted delivery of niclosamide, nanoparticles were functionalized with targeting ligands that correspond to certain overexpressed receptors on cancerous cells [80]. The incorporation of targeting ligands into niclosamide-loaded nanoparticle vehicles facilitates increased tumor selectivity while minimizing undesirable effects in peripheral tissues (Table 6). Proposed example ligands are folic acid, peptides, hyaluronic acid, carbohydrates, antibodies, and aptamers [81,82]. Some of these ligands were successfully used as conjugates to improve delivery in vitro and in vivo. There are currently several of these formulations being investigated in clinical trials [83,84,85,86]. One such targeting ligand is an antibody conjugation fragment aimed at targeting HER2 and EGFR, which are overexpressed in several breast and lung cancers [84].

It is also important to note that while the incorporation of ligands into nanotechnology allows for tumor-specific targeting, it may also result in “on-target, off-tumor” effects, wherein the vehicle’s payload is deposited in tissues that naturally express the antigen selected for targeting [86]. For example, this action was observed in targeted treatments against carbonic anhydrase in renal cell carcinoma, consequently leading to hepatotoxicity due to physiologic carbonic anhydrase expression in bile duct epithelium [94].

### 6.2. PEGylation

In addition to targeting specific tumor types, enhancing the circulation time of drug delivery systems can be achieved through surface modification with polyethylene glycol (PEG) [95]. PEGylation of surfaces hinders recognition and clearance of nanoparticles by the reticuloendothelial system, which is part of the immune system responsible for identifying and eliminating foreign substances [95]. This reduced recognition prolongs blood circulation time, increasing the chance of drug delivery to its target [95]. PEGylation of surfaces also facilitates increased biocompatibility and cytoplasmic transport. Researchers utilized targeted and PEG-coated formulations to enhance niclosamide’s efficacy, particularly in MCF7 breast cancer cells [95]. Eskandani et al. demonstrated increased incidence of apoptosis in MCF7 breast cancer cells using MUC1 aptamer (Ap)-targeted PLGA/PEGylated nanoparticles containing niclosamide when compared to a non-targeting formulation (76% in 72 h vs. 51%), highlighting the potential benefits of this approach in improving therapeutic outcomes [96].

Nanoparticle-based drug delivery systems showed many advantages over traditional drug delivery methods due to the vast protentional for modification. Future research challenges include the production of monodisperse particulate size, altering surface properties, and improvements surrounding drug loading capacity. Advanced materials and engineering strategies will be integral in refining the performance and biocompatibility of niclosamide-particle delivery systems in the future.

### 6.3. Electrospray Technique

Electrospraying is a technique where fine, homogenously charged, micro and nano liquid droplets are produced after the base liquid is subjected to an external electric field created by a sufficiently large voltage (several kilovolts) [97]. This electric field counters surface tension forces, leading to the jettison of fine droplets, which are guided by gases to a grounded collector [97]. Here, the droplets evaporate, and various micro/nano particulates of solute remain [97]. The size, structure, and morphology of the particulates are dependent on the structure of the droplet, which carries it to the collector and may be altered by changing the strength of the electric field [97]. This method allows for the consistent creation of a variety of solute morphologies, which result in changes to a solute’s absorption and diffusing properties [97]. Through the utilization of electrospraying and a novel suspension formulation, Lin et al. were able to successfully increase niclosamide’s *F* (25%) in rats and cytotoxicity in ovarian cancer lineages CP70 (IC50 3.59 µM) and SKOV3 (IC50 3.38 µM) in vitro. They also observed reduced tumor growth using these same lineages in a xenograft study in non-obese diabetic/severe combined immuno-deficient mice with their formulation [98]. They found no evidence of drug toxicity after investigating renal, hepatic, hematogenous, and general nutritional biomarkers, as well as brain, hepatic, renal, and intestinal histological samples [98].

### 6.4. Supercritical Technologies

Supercritical fluids are substances that share intermediate properties of gases and liquids, allowing for increased levels of diffusion while maintaining density [99]. The diffusion coefficient of a supercritical substance is over 10-fold greater than the diffusion coefficient of its liquid state [99]. The supercritical state is achieved when a substance is heated and pressurized beyond its critical point on its respective phase diagram [100]. Supercritical CO_2_ (scCO_2_) is a commonly used solvent in supercritical solutions and has FDA approval for pharmacological use [100]. This is in part due to its accessible critical point (temperature: 31.1 °C and pressure: 7.38 MPa) and benign and inert properties [101]. Through the utilization of a scCO_2_-solute solution, drugs may be loaded into vehicles, and then conditions change where the solution is no longer supercritical, leaving behind vehicle-loaded solute [101]. Many drugs, including niclosamide, were loaded into hydrophilic silica aerogels without drug degradation using this method [102]. This leads to improved drug absorption, attributed to increased drug surface area and favorable drug morphology compared to its crystalline form [102]. Niclosamide was shown to have an extremely poor affinity (0.01% loading = grams drug/grams aerogel) for hydrophilic silica aerogels using scCO_2_ as a solvent under conditions of 18 MPa and 40 °C [102]. The researchers suggest that niclosamide loading could be improved by increasing drug solubility in CO_2_ using entrainers like ethanol or acetone, or by using higher pressures during the loading phase [102].

## 7. Ongoing Clinical Trials of Niclosamide

### 7.1. Reformulated Niclosamide with Abiraterone/Prednisone with Castration-Resistant Prostate Cancer: NCT02519582

This phase Ib clinical trial assessed oral niclosamide’s safety and efficacy in metastatic prostatic cancer treatment [103]. In this initial trial, a combination therapy–oral niclosamide with abiraterone acetate, a CYP17A1 enzyme inhibitor (critical for decreasing androgen production) was tested. This study showed no increased efficacy of the combined therapy compared to the dosage of abiraterone acetate alone. The next phase will test a more orally bioavailable niclosamide derivative, PDMX1001 (niclosamide acetate, a prodrug of niclosamide) combined with abiraterone and prednisone. The outcome will help define the safety and tolerability of newly formulated niclosamide, as well as its potential for enhancing the clinical efficacy of abiraterone acetate

### 7.2. Niclosamide on Colon Cancer, Post-Primary Tumor Resection: ClinicalTrials.gov Identifier: NCT02687009

The pharmacokinetics and efficacy of unaltered oral niclosamide as a treatment for colon cancer are being evaluated in this trial [104]. An escalating dosage will be used to assess niclosamide’s effects on the dysregulated Wnt/β-catenin and NF-κB signaling pathways in colon cancer cells. Outcome measures include pharmacokinetic parameters, safety, and exploratory efficacy outcomes, such as response rates and changes in biomarkers associated with these pathways [104]

### 7.3. Combination Therapy of Niclosamide and Enzalutamide with Castration-Resistant Prostate Cancer: ClinicalTrials.gov Identifier: NCT04296851

The efficacy of combination therapy with oral niclosamide and enzalutamide, a nonsteroidal anti-androgen, was evaluated for the treatment of metastatic castration-resistant prostate cancer (mCRPC) [105]. Niclosamide previously demonstrated anti-cancer activity in preclinical studies on castration-resistant cancer cells [105]. This trial aimed to determine whether niclosamide could compensate for the drug resistance that typically develops against enzalutamide over the course of treatment. Efficacy was assessed through measures such as prostate-specific antigen (PSA) levels, progression-free survival, and overall survival rates. The trial revealed niclosamide’s poor bioavailability. Despite administering higher doses (1000 mg TID and 500 mg TID, compared to doses used for helminthic infections), plasma concentrations of niclosamide in all three patients ranged from 35.7 to 83 ng/mL, failing to reach the minimum effective concentration. Additionally, no decline in PSA levels was observed, leading the Data Safety Monitoring Board to close the trial.

## 8. Future Directions and Challenges

### 8.1. Bioavailability and Solubility Challenges

The successful preclinical assay outcomes of niclosamide are overshadowed by challenges pertaining to a lack of bioavailability and solubility, both of which limit its potential as an anti-cancer agent. Bioavailability enhancements were approached by generating chemical derivatives and exploring alternative drug delivery systems. For example, studies examining niclosamide ethanolamine (NEN) indicated potential solubility improvement and reduced hepatic metastasis in colorectal cancer animal models [5]. In addition, researchers have begun examining nanoparticle-based delivery systems to enhance the cellular uptake of niclosamide and initiate accumulation within tumor tissue. Future works will ultimately need to hone these delivery systems to enhance absorption throughout the body and direct delivery to tumor tissue sites where a direct clinical impact might be possible.

### 8.2. Mechanisms of Resistance

It is important to note that cancer cells acquire resistance mechanisms to attenuate and neutralize the effects of anti-neoplastic agents. Tumor acquisition of niclosamide resistance mechanisms include decreased drug uptake, enhanced efflux, and changes in metabolism [10]. Ultimately, the identification of resistance mechanisms in target cancers can guide the design of combination therapy, wherein appropriate drugs may be employed synergistically to retain therapeutic efficacy by overcoming or bypassing resistance [10,54]. For instance, some research has successfully employed niclosamide in tandem with other cancer-targeting agents, chemotherapies, and immune checkpoint inhibitors to increase sensitivity to niclosamide and/or reduce resistance. Investigating potential resistance pathways or mutations associated with niclosamide resistance may assist in developing innovative combinations that address tumor resistance and improve clinical efficacy.

## 9. Conclusions

Niclosamide shows great promise as an anticancer agent due to its action in multiple cancer pathways. To surmount the challenges associated with bioavailability and solubility, continued development of new delivery systems and chemical modifications of niclosamide will be crucial. Ongoing and future preclinical and clinical studies will shed more light on niclosamide’s potential use as an anti-cancer treatment, including its mechanisms of action, patterns of resistance, and usefulness in combination with other therapies. There is potential for new applications and repurposing of the drug, leading to more effective and personalized treatment regimens, providing clinicians and researchers with a novel tool in their fight against cancer.

## Figures and Tables

**Figure 1 cancers-16-03548-f001:**
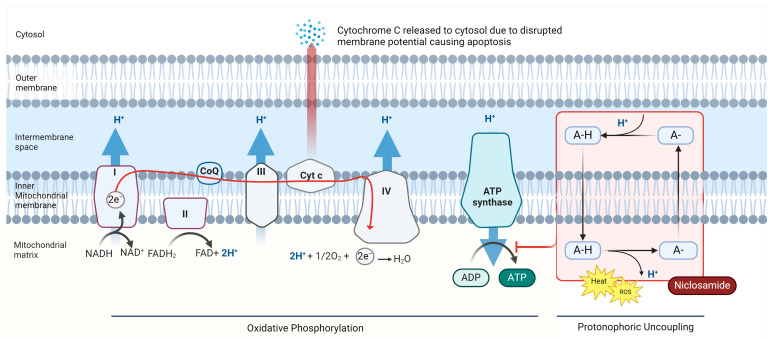
Niclosamide’s action as a mitochondrial uncoupler and protonophore. Mitochondrial membrane depolarization is believed to lead to mitochondrial matrix condensation and subsequent Cyt*c* release, an initiator of apoptosis. Niclosamide also contributes to the generation of reactive oxygen species and activation of AMP-activated protein kinase (AMPK) due to inhibition of ATP synthesis. The accumulation of these effects is believed to be a major mechanism of niclosamide-induced tumor cell death (created with Bio Render) [19,20,21].

**Figure 2 cancers-16-03548-f002:**
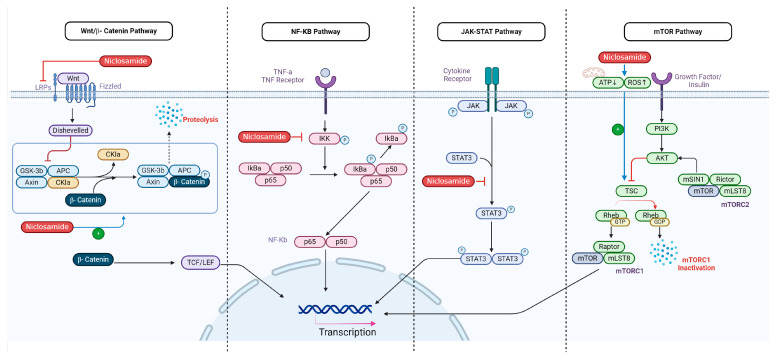
Illustrates the multifaceted mechanisms of action of niclosamide in cancer treatment. It depicts niclosamide as a mitochondrial uncoupler that disrupts ATP production and induces Cytc release, leading to apoptosis through reactive oxygen species (ROS) generation, and niclosamide’s inhibition of the STAT3 signaling pathway, blocking cancer cell proliferation; disruption of the mTOR pathway via activation of the tuberous sclerosis complex (TSC); degradation of LRP6 to inhibit the Wnt/β-catenin pathway; and suppression of the NF-κB pathway by inhibiting IκB kinase (IKK), thus reducing anti-apoptotic gene expression [32]. Collectively, these actions highlight niclosamide’s potential as a versatile therapeutic agent in oncology (Created with BioRender).

**Figure 3 cancers-16-03548-f003:**
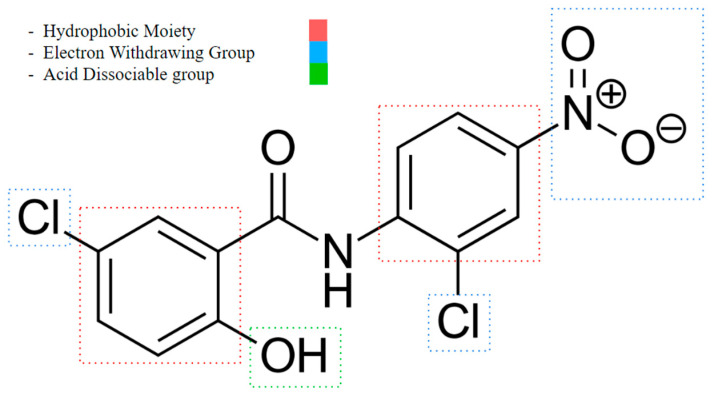
Niclosamide’s functional group characteristics required it to act as an uncoupler, noting its hydrophobic moiety, electron-withdrawing group, and acid-dissociable group.

**Table 1 cancers-16-03548-t001:** Comparison of niclosamide and its derivative’s solubility in water.

Niclosamide and Derivatives	Solubility in Water, 20~25 °C
Niclosamide	0.0016 g/L [9]
Niclosamide Ethanolamine Salt	~21 g/L [10]
Niclosamide Piperazine Salt	~30 g/L [10]
Niclosamide Nicotinamide Co-crystals	~0.009 g/L [11]
Niclosamide Octenylsuccinate Hydroxypropyl Phytoglycogen (OHPP)	0.133 g/L [12]

**Table 2 cancers-16-03548-t002:** Efficacy of niclosamide and its derivatives on glioblastoma cells [73].

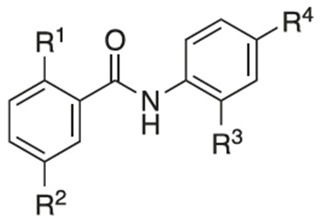					
Compound	R^1^	R^2^	R^3^	R^4^	IC50 (µM)
Niclosamide	OH	Cl	Cl	NO_2_	1.54
**2**	OH	Cl	H	NO_2_	1.58
**5**	OH	Cl	H	NH_2_	12.77
**6**	OH	Cl	Cl	N_3_	1.52
**7**	OH	Cl	Cl	CF_3_	1.55
**13**	OH	H	Cl	NO_2_	1.88

**Table 3 cancers-16-03548-t003:** Efficacy of niclosamide and its derivatives on MDA-MB-231 xenograft tumor growth in vivo [74].

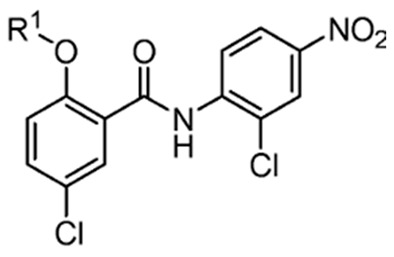					
Compound	R^1^	IC_50_ (μM)			
		Breast Cancer		Pancreatic Cancer	
		MCF-7	MDA-MB-231	AsPC1	Panc-1
Niclosamide		1.06	0.79	1.47	1.73
**10**	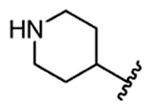	0.25	0.29	2.76	0.54
**11**	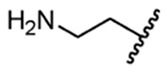	0.91	1.64	1.9	1.08

**Table 4 cancers-16-03548-t004:** Efficacy of niclosamide derivatives on three cancer cell lines and one normal cell line in vitro [75].

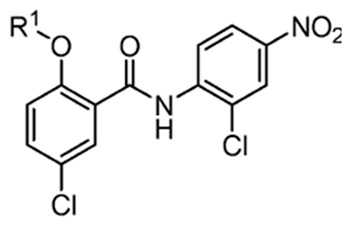					
Compound	R^1^	IC_50_(μM)			
		A549	NCl-H446	Jurkat	HBE (Normal Cell Line)
**1**	H	7.70	0.96	1.8	6.42
**2**	COCH_3_	4.13	0.92	1.5	5.76

**Table 5 cancers-16-03548-t005:** Efficacy of niclosamide derivatives in the Frizzled1-GFP internalization and the TOPflash β-β-catenin reporter assays [76].

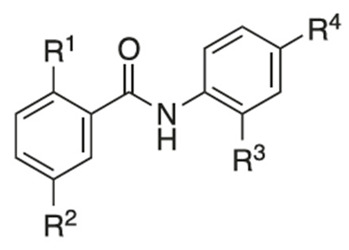					
Compound	R^1^	R^2^	R^3^	R^4^	IC50 (µM)
Niclosamide	OH	Cl	Cl	NO_2_	0.4
**4**	OH	Br	Cl	NO_2_	0.26
**5**	OH	Br	H	CF_3_	1.2
**6**	OH	Cl	H	NO_2_	1.2
**7**	OH	Cl	OCH_3_	NO_2_	0.44

**Table 6 cancers-16-03548-t006:** Cytotoxic activity of niclosamide nanoparticle-based delivery formulations in cancer cell lines of different tissue origin.

Nanoparticle	Target Cells	IC_50_ (μM) Nano-Niclosamide	IC_50_ (μM) Niclosamide	Nanoparticle Size (nm)
Elastin-like polypeptide	Colon cancer (*HCT116*)	0.94 µM [80]	0.85	~74 nm
PEO, Ag poly(e-caprolactone)	Lung cancer (*A549*) Breast cancer (*MCF-7*)	1.24 µM [87] 1.21 µM [87]	1.45 µM 6.5 µg/mL	632 nm
Pluronic^®^, biotin	Lung cancer (*A549*)	<0.3 µM [88]	0.9 µM	25–35 nm
Polydopamine Nanoparticles	Breast Cancer (MDA-MB-231)	2.73 µM [89]	1.88 µM	146.3 nm
Nanocarbon	Breast Cancer (MCF-7)	21 µM [90]	45 µM	~55 nm
Stearate Prodrug	Osteosarcoma (143B, MG63, U2OS, and SaOS2)	1.27 µM [91]	1.16 μM	~100 nm
Chitosan Nanoparticles	Lung Cancer (A549), Breast Cancer (MCF-7)	8.75 µM, 7.5 µM [92]	Not provided	100–120 nm
Polypeptide Nanoparticles	Colorectal Cancer (HCT116)	0.94 µM [93]	0.85	<100 nm

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
