# Peer review of "Optimizing Niclosamide for Cancer Therapy: Improving Bioavailability via Structural Modification and Nanotechnology"

_cancers, 2024, doi:10.3390/cancers16203548_

Round 1
Reviewer 1 Report
Comments and Suggestions for Authors
The article is devoted to the drug niclosamide, its antitumor activity and mechanisms of its action on tumor cells, as well as possible ways to overcome its low activity due to low solubility. Various possibilities for increasing the drug's effectiveness are considered, from converting it to a nanoform and including it in targeted delivery systems to using structurally similar compounds. The presented material is relevant and interesting for researchers in the field of antitumor drugs: it describes in detail the effect of niclosamide (previously used to treat helminths) on tumor cells with references to a large number of sources. The review may also be of interest to researchers involved in increasing the effectiveness of poorly soluble drugs. However, when reading, questions arise both about the structure of the article and its title.
- Part 2, devoted to the drug's effect on cells and mechanisms, has a very strange structure. Parts 2.1.1. and 2.2.1. are unsuccessfully separated from parts 2.1 and 2.2. - this is confusing (parts 2.1.2. and 2.2.2. are missing and this principle of dividing information is not used in parts 2.3, 2.4, 2.5). It is necessary to either include parts 2.1.1. and 2.2.1. in 2.1 and 2.2. or divide part 2 into subparts differently.
- Possible related compounds with better solubility that can serve as a replacement for this drug, criteria for their selection are proposed. References to articles devoted to testing these compounds are given. But this can be attributed to chemical modification with great difficulty, and not in all cases (and then reactions of converting niclosamide into these compounds are needed). Or the question of changing the review title arises.
- Physical methods of increasing the effectiveness of this compound (without changing the chemical composition, but due to micro- and nanonization, conversion into suspensions) are considered slightly. This again raises the question of the appropriateness of the title. The article lacks information on the use of Electro sprayed technique, Supercritical technologies, Cryogenic technologies for nano- and micronization, transfer to suspension form. On the growth of bioavailability of nanoforms due to high specific surface area and increased saturation solubility.
- Considering the title, more information can be added on the effectiveness of targeted delivery systems (it is desirable to describe in more detail)
Author Response
We sincerely thank you for reviewing this paper and for your feedback. We are pleased to note your comments and have made the following corrections in this revision.
The article is devoted to the drug niclosamide, its antitumor activity and mechanisms of its action on tumor cells, as well as possible ways to overcome its low activity due to low solubility. Various possibilities for increasing the drug's effectiveness are considered, from converting it to a nanoform and including it in targeted delivery systems to using structurally similar compounds. The presented material is relevant and interesting for researchers in the field of antitumor drugs: it describes in detail the effect of niclosamide (previously used to treat helminths) on tumor cells with references to a large number of sources. The review may also be of interest to researchers involved in increasing the effectiveness of poorly soluble drugs. However, when reading, questions arise both about the structure of the article and its title.
=> Restructured the article: 1) Reduced mechanistic section. 2) Added additional nanoparticle-based delivery system 3) Removed Lipid section, and 4) Changed the title accordingly to "A Review of Niclosamide Chemical Derivatives and Nanoparticle Delivery Systems for Enhanced Cancer Therapy."
- Part 2, devoted to the drug's effect on cells and mechanisms, has a very strange structure. Parts 2.1.1. and 2.2.1. are unsuccessfully separated from parts 2.1 and 2.2. - this is confusing (parts 2.1.2. and 2.2.2. are missing and this principle of dividing information is not used in parts 2.3, 2.4, 2.5). It is necessary to either include parts 2.1.1. and 2.2.1. in 2.1 and 2.2. or divide part 2 into subparts differently.
=> 2.2.1, 2.1.1 and similar sections are reorganized to a separate section
- Possible related compounds with better solubility that can serve as a replacement for this drug, criteria for their selection are proposed. References to articles devoted to testing these compounds are given. But this can be attributed to chemical modification with great difficulty, and not in all cases (and then reactions of converting niclosamide into these compounds are needed). Or the question of changing the review title arises.
=> changed the title: A Review of Niclosamide Chemical Derivatives and Nanoparticle Delivery Systems for Enhanced Cancer Therapy.
- Physical methods of increasing the effectiveness of this compound (without changing the chemical composition, but due to micro- and nanonization, conversion into suspensions) are considered slightly. This again raises the question of the appropriateness of the title. The article lacks information on the use of Electro sprayed technique, Supercritical technologies, Cryogenic technologies for nano- and micronization, transfer to suspension form. On the growth of bioavailability of nanoforms due to high specific surface area and increased saturation solubility.
=>3.2.5 Micellar Formulations for Niclosamide &. 3.2.4 Liposomal and Micellar Formulations for Enhanced Delivery of Niclosamide removed. Also added Electro sprayed technique, Supercritical technologies, Cryogenic technologies for nano- and micronization Transfer to Suspension Form Nanoforms
- Considering the title, more information can be added on the effectiveness of targeted delivery systems (it is desirable to describe in more detail)
=> Added more targeted delivery systems and hanged the title: A Review of Niclosamide Chemical Derivatives and Nanoparticle Delivery Systems for Enhanced Cancer Therapy
Reviewer 2 Report
Comments and Suggestions for Authors
General comment: Please expand the abbreviations the first time they appear in the manuscript.
Line 10: Remove the capital letter in “Niclosamide/Niclosamide Ethanolamine (NEN)/Niclosamide Piperazine (NPP)” throughout the manuscript.
Line 43: Please provide the full names of the abbreviations the first time they appear in the text.
Line 83: Replace “Niclosamide’s action as a mitochondrial uncoupler and protonophore” with “Niclosamide’s works as/functions as”
Figure 1 description: Please provide the full names of the abbreviations used in the figure and its description.
Line 126: Replace “ higher IC50” with “higher concentration”
Line 131: Replace “selectively” with “selectivity”
Figure 2 description should be provided.
Line 184: Remove “negative” from “negative inhibition”
Line 213: Should be “It was also observed that the apoptosis induction was increased….”
Line 225: Promoting degradation of what? Please specify.
Line 233: Replace “would decrease” to “is decreased” or “decrease”
Line 374-381 and 410-413: I suggest rewriting these parts as they are not easily understandable.
Table 6. I suggest changing the table title to: “Cytotoxic activity of niclosamide nanoparticle-based delivery formulations in cancer cell lines of different tissue origin.”
I suggest rewriting: “Targeting ligands incorporated into Niclosamide-loaded nanoparticle vehicles offer potential to increase tumor selectivity and while minimizing undesirable effects in peripheral tissues” to “Incorporation of targeting ligands into niclosamide-loaded nanoparticle vehicles offers the potential to increase tumor selectivity and minimize undesirable effects in peripheral tissues”
Line 423: I suggest replacing “many remain in clinical trials” with “many are investigated in clinical trials”
Line 423-425: I suggest replacing “Overexpression of receptors and tumor-associated antigens on tumor cell surfaces offer prime targeting by smart nanotechnologies” with “Overexpression of receptors and tumor-associated antigens on tumor cell surfaces offer opportunities for prime targeting of these cells by smart nanotechnologies”
Line 425: I suggest replacing “conjugated” with “embedded in the nanoparticle structure”
Line 431: I suggest replacing: “One should be mindful that” to “It is also important, that while…”
Line 440-443: This fragment should be rewritten as it is not clear in the current form.
Line 447: Remove “so”
Line 454: I suggest replacing “evolving” with some more appropriate word in this context.
Line 459-480: English editing for this part is required. I also suggest re-ordering information appearing in this paragraph to enhance clarity.
Line 484: Replace “Several studies developed” with “Several micellar formulations using Pluronic F12 have been developed”. Re-structure the sentence to continue writing. For example, do not try to incorporate too much information in a single sentence. Rather, divide the sentences by providing detailed information on specific parts.
Line 486: replace “When gathered” with “Following assembly”
Line 492: replace “decorated” with a more appropriate word.
Section 3.3. need to be restructured. The headings must not contain the reference. Also, the text provided in the main text of each subparagraph looks like an unverified paraphrase of the text provided in clinicaltrails.gov. Authors should rewrite this part.
Lines 550-553: “For example, studies examining Niclosamide Ethanolamine (NEN) indicated a potential for solubility improvement and reduced hepatic metastasis in colorectal cancer animal models, among other improvements when the chemotherapeutic, Niclosamide, was chemically modified” here is also an example of unsupervised paraphrase using software like quillbot. It needs to be rewritten according to academic standards of writing.
Line 565: replace “inform” with a more appropriate word.
Line 577: replace “ligatures” with a more appropriate word.
Comments on the Quality of English LanguageModerate editing of English language required.
Author Response
We sincerely thank you for reviewing this paper and for your feedback. We are pleased to note your comments and have made the following corrections in this revision.
General comment: Please expand the abbreviations the first time they appear in the manuscript.
=> Added
Line 10: Remove the capital letter in “Niclosamide/Niclosamide Ethanolamine (NEN)/Niclosamide Piperazine (NPP)” throughout the manuscript.
=> Removed
Line 43: Please provide the full names of the abbreviations the first time they appear in the text.
=> Provided
Line 83: Replace “Niclosamide’s action as a mitochondrial uncoupler and protonophore” with “Niclosamide’s works as/functions as”
=> Sentence unnecessary. Deleted.
Figure 1 description: Please provide the full names of the abbreviations used in the figure and its description.
=> Provided. Left CoQ as is as it’s common nomenclature.
Line 126: Replace “ higher IC50” with “higher concentration”
=> Replaced
Line 131: Replace “selectively” with “selectivity”
=> Replaced
Figure 2 description should be provided.
=> Provided
Line 184: Remove “negative” from “negative inhibition”
=> Removed
Line 213: Should be “It was also observed that the apoptosis induction was increased….”
=> Changed
Line 225: Promoting degradation of what? Please specify.
=> According to another reviewer's comment, we reorganized this section.
Line 233: Replace “would decrease” to “is decreased” or “decrease”
=> According to another reviewer's comment, we reorganized this section.
Line 374-381 and 410-413: I suggest rewriting these parts as they are not easily understandable.
=> Rewritten
Table 6. I suggest changing the table title to: “Cytotoxic activity of niclosamide nanoparticle-based delivery formulations in cancer cell lines of different tissue origin.”
=> Changed
I suggest rewriting: “Targeting ligands incorporated into Niclosamide-loaded nanoparticle vehicles offer potential to increase tumor selectivity and while minimizing undesirable effects in peripheral tissues” to “Incorporation of targeting ligands into niclosamide-loaded nanoparticle vehicles offers the potential to increase tumor selectivity and minimize undesirable effects in peripheral tissues”
=> Replaced as suggested
Line 423: I suggest replacing “many remain in clinical trials” with “many are investigated in clinical trials”
=> Replaced
Line 423-425: I suggest replacing “Overexpression of receptors and tumor-associated antigens on tumor cell surfaces offer prime targeting by smart nanotechnologies” with “Overexpression of receptors and tumor-associated antigens on tumor cell surfaces offer opportunities for prime targeting of these cells by smart nanotechnologies”
=> Replaced
Line 425: I suggest replacing “conjugated” with “embedded in the nanoparticle structure”
=> Replaced
Line 431: I suggest replacing: “One should be mindful that” to “It is also important, that while…”
=> Replaced
Line 440-443: This fragment should be rewritten as it is not clear in the current form.
=> Rewritten
Line 447: Remove “so”
=> Removed
Line 454: I suggest replacing “evolving” with some more appropriate word in this context.
=> Replaced
Line 459-480: English editing for this part is required. I also suggest re-ordering information appearing in this paragraph to enhance clarity.
=> Other reviewers suggested to remove this section. To avoid any confusion, the liposomal section was removed
Line 484: Replace “Several studies developed” with “Several micellar formulations using Pluronic F12 have been developed”. Re-structure the sentence to continue writing. For example, do not try to incorporate too much information in a single sentence. Rather, divide the sentences by providing detailed information on specific parts.
=> Liposomal section was removed
Line 486: replace “When gathered” with “Following assembly”
=> Liposomal section was removed
Line 492: replace “decorated” with a more appropriate word.
=> Liposomal section was removed
Section 3.3. need to be restructured. The headings must not contain the reference. Also, the text provided in the main text of each subparagraph looks like an unverified paraphrase of the text provided in clinicaltrails.gov. Authors should rewrite this part.
=> Reference in the heading removed. Section 3 was significantly restructured for clarity.
Lines 550-553: “For example, studies examining Niclosamide Ethanolamine (NEN) indicated a potential for solubility improvement and reduced hepatic metastasis in colorectal cancer animal models, among other improvements when the chemotherapeutic, Niclosamide, was chemically modified” here is also an example of unsupervised paraphrase using software like quillbot. It needs to be rewritten according to academic standards of writing.
=> Paragraph rewritten “For example, studies examining niclosamide ethanolamine (NEN) indicated potential solubility improvement and reduced hepatic metastasis in colorectal cancer animal models.” (Line 600)
Line 565: replace “inform” with a more appropriate word.
=> Sentence rewritten "Ultimately, the identification of resistance mechanisms in target cancers can guide the design of combination therapy" (Line 615)
Line 577: replace “ligatures” with a more appropriate word.
=> Sentence rewritten "Niclosamide shows great promise as an anticancer agent due to its action in multiple cancer pathways." (Line 624)
Reviewer 3 Report
Comments and Suggestions for Authors
Dear Authors,
This review paper tries to provide a comprehensive literature search on improving Niclosamide’s effectiveness in cancer therapy: Advances in drug modification and delivery. It seems quite interesting to the readers. However, the following bullet points need to be enhanced to make its contribution to the literature better.
1. In the abstract section, the authors described that advanced drug delivery methods, particularly nanotechnology make drugs better in regards to drug delivery features. However, they haven’t written names of them throughout the manuscript what are those methods, and compare their outputs for each of them.
2. In the abstract section, the authors informed that this review includes fewer side effects of advanced drug delivery techniques-based drugs on cancer therapy but they have not written what could be them in the manuscript.
3. In the abstract, the authors named two different derivatives of Niclosamide in the manuscript which are NPP (Niclosamide piperazine) and NEN (Niclosamide ethanolamine). However, they described only NEN and gave literature about it. They should also provide literature on NPP in the relevant part of the manuscript.
4. On page 3 of 20 line 83, the authors should check the sentence in grammar and make corrections on it. They also need to double-check others to polish similar ones.
5. On page 9 line 341, Table 2 and 3 describe some sort of Niclosamide derivatives by using compound labels such as 2, 5, 6, 7, 10, and 13. However, the formulations were not written in the manuscript or table captions which appears blurry area for the readers. They should add the formulations for each compound.
6. The authors described different nanotechnology techniques to produce nanostructures for advanced drug delivery of niclosamide and its derivatives. But, they designed this section with the literature of niclosamide incorporated nanoparticles which seems this section presents a lack of information in this area. They should put literature on nanocrystals, nanofibers, and carbon particles related to Niclosamide and its derivative-centered research works into the manuscript.
Kind regards,
Author Response
We sincerely thank you for reviewing this paper and for your feedback. We are pleased to note your comments and have made the following corrections in this revision.
- In the abstract section, the authors described that advanced drug delivery methods, particularly nanotechnology make drugs better in regards to drug delivery features. However, they haven’t written names of them throughout the manuscript, what are those methods, and compare their outputs for each of them.
=> abstract was rewritten
- In the abstract section, the authors informed that this review includes fewer side effects of advanced drug delivery techniques-based drugs on cancer therapy but they have not written what could be them in the manuscript.
=> abstract updated to remove mentioning of side effects
- In the abstract, the authors named two different derivatives of Niclosamide in the manuscript which are NPP (Niclosamide piperazine) and NEN (Niclosamide ethanolamine). However, they described only NEN and gave literature about it. They should also provide literature on NPP in the relevant part of the manuscript.
=> Added NPP section (Section 5.2)
- On page 3 of 20 line 83, the authors should check the sentence in grammar and make corrections on it. They also need to double-check others to polish similar ones.
=> Checked grammatical errors
- On page 9 line 341, Table 2 and 3 describe some sort of Niclosamide derivatives by using compound labels such as 2, 5, 6, 7, 10, and 13. However, the formulations were not written in the manuscript or table captions which appears blurry area for the readers. They should add the formulations for each compound.
=> Formulation (synthesis part) was too lengthy, and hindered flow. Thus, we do not include the information. Instead, re-named it based on the functional group modification, following the convention of the original sources. Numbering of the compound in the Table is from the original authors' compound labeling. The original authors did not provide chemical compound name, and therefore, we thought it would be better to call out author’s naming convention & note the changed functional group so that the readers can refer back easily to the original source for further review, and see the difference from the original niclosamide.
- The authors described different nanotechnology techniques to produce nanostructures for advanced drug delivery of niclosamide and its derivatives. But, they designed this section with the literature of niclosamide incorporated nanoparticles which seems this section presents a lack of information in this area. They should put literature on nanocrystals, nanofibers, and carbon particles related to Niclosamide and its derivative-centered research works into the manuscript.
=> added more section on this (Section 6) and deleted liposomal delivery to focus
Round 2
Reviewer 1 Report
Comments and Suggestions for Authors
The authors have worked on the text. The title already matches the content. The parts on nanotechnology have been added and the oddities in the structure have been removed. Remarkable work has been done. It can be approved for publication.
Minor
Line 125: "aandniclosamide’s" - misprint "and niclosamide’s"
Line 245: "bortezomib is effective in many malignancies" - better "bortezomib is effective against many malignancies"
Line 336: "to niclosamide These hydrophobic" - period missing "to niclosamide. These hydrophobic"
Author Response
We truly appreciate your time to review the revision and provide feedback. We made the following corrections accordingly. The changed parts are highlighted in green. (The first revision was yellow).
Minor
Line 125: "aandniclosamide’s" - misprint "and niclosamide’s"
- Corrected
Line 245: "bortezomib is effective in many malignancies" - better "bortezomib is effective against many malignancies"
- Corrected
Line 366: "to niclosamide These hydrophobic" - period missing "to niclosamide. These hydrophobic"
- Added
Reviewer 2 Report
Comments and Suggestions for Authors
The authors have generally corrected the paper according to my recommendations. Nevertheless, they have not paid enough attention to the details as:
a) some abbreviations are still not expanded first time they appear in the text as STAT3, NF-κB, Wnt/β-catenin, mTOR (line 43),
b) Figure 1 description - AMPK still lacks full name,
c) Despite providing Figure 2 description authors failed to follow remarks from the other comments as abbreviations are not expanded and Niclosamide begins with capital N (please go through the whole manuscript as it is still mixed).
After the changes are introduced, I recommend it for acceptance in the revised form.
Minor editing of English language required.
Author Response
We truly appreciate the time you took to review the revision and give us feedback. We apologize for missing some corrections you provided in the first round. We made the following corrections accordingly. The changed parts are highlighted in green.
a) The authors have generally corrected the paper according to my recommendations. Nevertheless, they have not paid enough attention to the details as:
a) some abbreviations are still not expanded first time they appear in the text as STAT3, NF-κB, Wnt/β-catenin, mTOR (line 43),
- Expanded to “such as signal transducer and activator of transcription 3 (STAT3), nuclear factor-kappa B (NF-κB), wingless-related integration site (Wnt)/β-catenin, mammalian target of rapamycin (mTOR),”
b) Figure 1 description - AMPK still lacks full name,
- Expanded to “AMP-activated protein kinase (AMPK)
c) Despite providing Figure 2 description authors failed to follow remarks from the other comments as abbreviations are not expanded and Niclosamide begins with capital N (please go through the whole manuscript as it is still mixed).
- Expanded the abbreviations TSC to tuberous sclerosis complex (TSC) in Figure 2. (Line 129). The other abbreviations are mentioned before Figure 2.
- Changed Niclodamide to niclosamide when it is not the beginning of a sentence. (Lines 123, 124, 129, 207, 241, 436, 442, 472, 574, 590)
In addition, we read through the whole manuscript and changed the following parts
- Line 147: phosphoinositide 3-kinases (PI3K), protein kinase B (Akt), and phosphatase and tensin homolog (PTEN)
- Line 170: T-cell factor (TCF) and lymphoid enhancer factor (LEF)
- Line 173: cellular myelocytomatosis (c-Myc), vascular endothelial growth factor (VEGF), matrix metalloproteinase 7 (MMP7),
Reviewer 3 Report
Comments and Suggestions for Authors
The authors have adequately addressed my comments in this revised version. Therefore, I have no further comments.
Author Response
We truly appreciate your time to review the revision and to approve the revised version.